# On the impact of capillarity for strength at the nanoscale

Nadiia Mameka[1], Jürgen Markmann [1,2] & Jörg Weissmüller [1,2]

The interior of nanoscale crystals experiences stress that compensates for the capillary forces and that can be large, in the order of 1 GPa. Various studies have speculated on whether and how this surface-induced stress affects the stability and plasticity of small crystals. Yet, experiments have so far failed to discriminate between the surface contribution and other, bulk-related size effects. To clarify the issue, here we study the variation of the flow stress of a nanomaterial while distinctly different variations of the two capillary parameters, surface tension, and surface stress, are imposed under control of an applied electric potential. Our theory qualifies the suggested impact of surface stress as not forceful and instead predicts a significant contribution of the surface energy, as measured by the surface tension. The predictions for the combined potential-dependence and size-dependence of the flow stress are quantitatively supported by the experiment. Previous suggestions, favoring the surface stress as the relevant capillary parameter, are not consistent with our experiment.

[1] Institute of Materials Research, Materials Mechanics, Helmholtz-Zentrum Geesthacht, Max-Planck-Straße 1, 21502 Geesthacht, Germany. [2] Institute of Materials Physics and Technology, Hamburg University of Technology, Eissendorfer Straße 42, 21073 Hamburg, Germany. Correspondence and requests for materials should be addressed to J.W. (email: weissmueller@tuhh.de)

Even when there is no external load, the interior of a small crystal experiences a surface-induced stress, $\sigma_C$, which compensates the capillary forces that are quantified by the surface stress, $f$ (ref. [1]). With the magnitude of $\sigma_C$ in the order of $f/r$ (ref. [2]) and $f$ in the order of $3\,N\,m^{-1}$ (ref. [3]), stresses in excess of 1 GPa are expected for crystals with characteristic radius, $r$, at the lower nanoscale. As $\sigma_C$ typically has a significant deviatoric stress component[2], the capillary forces may affect the shear deformation of crystal plasticity. Indeed, an instability to spontaneous plastic deformation is observed in extremely small structures. Atomistic simulation studies[4–8] and, on a more speculative note, experimental reports[9,10] attribute the instability to plastic shear prompted by the action of the surface-induced stress. It has also been suggested that $\sigma_C$ will enhance the action of a compressive external load and diminish that of a tensile load, resulting in a substantial tension–compression asymmetry in the strength of nanowires[6,11–13] and in the strength[7,8,14] as well as creep rate[15] of nanoporous metals. Thus, surface stress is believed to impose a lower limit on the stable size of crystals and to contribute substantially to nanoscale mechanical behavior.

Yet, a separate capillary parameter has also been considered in the context of surface effects on crystal plasticity: The surface tension, $\gamma$, represents a specific excess free energy per area of surface. It is known that $\gamma$ may prompt spontaneous shortening of macroscopic metal wires by creep at elevated temperature. Zero-creep measurements, pioneered by H. Udin around 1950[16,17] and later extended to multilayers[18], measure $\gamma$ via the tensile load required to suppress the contraction. The impact of $\gamma$ on plasticity is further emphasized by studies of engineering materials wetted by electrolytes. These reveal similarities between the electrode-potential dependence of $\gamma$ and creep rate[19,20] or fracture stress[21]. Zero-creep experiments typically use wires a few tens of μm in diameter and very low stresses, in the order of 10–100 kPa. Yet, as the impact of surface phenomena is enhanced at small size, much larger surface-related stresses may be expected for nanowires.

The net surface excess free energy, $G_S$, scales with the surface area, $A$. Wires tend to contract spontaneously in zero-creep experiments as the contraction reduces $A$ and, thereby, the energy: $\delta G_S = \gamma \delta A$. The stress that is required for compensating the trend for contraction—resulting in zero-creep rate—in a wire of radius $r$ is tensile and of magnitude $\gamma/r$ (refs. [16,17]). Zero-creep experiments thus exemplify that tension–compression asymmetry results from the action of surface tension: creep is arrested by tensile stress but would be accelerated by a compressive stress of same magnitude. It is less obvious how the notion of a shear deformation driven by surface stress connects to energy minimization. In fact, plastic shear may create slip steps or terraces of new crystallographic orientation, both of which may in principle increase $G_S$, excluding a spontaneous process. Suggestions of surface stress as a driving force for spontaneous shear have so far not been linked to the energetics, and the present work addresses the issue.

Recent studies of the deformation of nanoporous gold (NPG) in situ in electrolyte present new opportunities for investigating nanoscale mechanical behavior by experiment. NPG is an emerging model nanomaterial that can be made with mm dimensions and tested using reliable macroscopic testing schemes[22]. The polycrystalline material with 10–100 μm grain size is distinguished by its network structure of nanoscale struts or ligaments. The brittle failure of NPG in tension relates to fracture mechanics concepts such as the distribution of heterogeneities in the network structure[23]. By contrast, the material's excellent deformability in compression provides opportunities for probing the mechanisms and driving forces of yielding and plastic flow in small-scale plasticity. In fact, the mechanical behavior of the ligaments agrees well with that of gold nanopillars and

nanowires[22,24–26], supporting the relevance of studies of NPG for understanding small-scale plasticity in general.

In situ tests of NPG in electrolyte allow monitoring the mechanical behavior, while simultaneously the surface state is modulated under control of the electrode potential, $E$ (refs. [14,27–29]). Quite recently, this approach has provided support for a tension–compression asymmetry in the plastic flow of NPG[14]. Yet, the nature of the underlying capillary force, surface stress of surface tension, remains to be studied. Here, we approach this issue, again using in situ tests of NPG in electrolyte. We analyze the relevant aspects of the mechanics of small crystals and we link this analysis to experiments in which the surface state is modulated during the deformation and the response of the flow stress monitored. Specifically, these experiments exploit that it is known how the capillary forces $\gamma$ and $f$ vary independently with $E$ (ref. [30]). Therefore, the response of the mechanical behavior to potential variation affords a distinction between the action of the two capillary parameters. The observations do not support the suggested impact of surface stress on strength. Instead, they agree quantitatively with the predicted action of the surface tension, which is not typically considered in this context.

## Results

**Variation of net surface energy during deformation.** Our analysis admits that a part, $\Delta T$, of the external applied stress, $T$, that drives the deformation is required for doing work against as-yet unspecified capillary forces. This part is not available for overcoming the intrinsic dissipative forces that determine the resistance to dislocation motion in bulk plasticity. The apparent experimental flow stress, $\sigma^{flow}$, is then the sum of the intrinsic bulk flow stress, $\sigma_0^{flow}$, and of $\Delta T$:

$$\sigma^{flow} = \sigma_0^{flow} + \Delta T. \qquad (1)$$

We here explore two possible origins of $\Delta T$.

We analyze a solid with volume $V$ and surface area $A$, which are both measured in stress-free states of the solid. The total free energy may be expressed as $G = V\Psi + A\gamma$ with $\Psi$ the volumetric free energy density in the bulk. As the defining equation for $\gamma$ we may thus take

$$\gamma = G_S/A = (G - V\Psi)/A, \qquad (2)$$

so that $\gamma$ is the excess, per area, in free energy over that of a bulk solid with same volume but negligible surface effects. If the plastic strain $\delta\varepsilon^p$ changes $A$ or $\gamma$, then a part, $\Delta T V \delta\varepsilon^p$, of the mechanical work is consumed for supplying the extra energy $\delta(\gamma A)$. Equating mechanical work and free energy change yields the required extra traction as

$$\Delta T = \frac{1}{V}\frac{\delta G_S}{\delta\varepsilon^p} = \gamma\frac{\delta\alpha}{\delta\varepsilon^p} + \alpha\frac{\delta\gamma}{\delta\varepsilon^p}, \qquad (3)$$

with $\alpha = A/V$ the volume-specific surface area. Contrary to dissipative processes of classic plasticity, the impact of surface tension on the flow stress links to a conservative process that stores or releases energy.

**Surface-induced stress and relaxation.** The capillary parameter that relates to elasticity is the surface stress, $f$. It quantifies the tendency of the surface to compress ($f > 0$) or expand ($f < 0$) the solid elastically. Restricting attention to isotropic surfaces, we take $f = d\gamma/de$ with $e$ the relative change in surface area (in laboratory coordinates) by tangential elastic strain.

As a model that incorporates the most obvious features of a small-scale solid, we consider the long (negligible end-effects) cylindrical nanowire of radius $r$ and length $l \gg r$, for which $\alpha =$

$2/r$. We take elasticity, surface tension, and surface stress as isotropic.

Even in the absence of an applied load, the surface stress requires a compensating stress $\sigma_C$ in the bulk of the nanowire; its axial and radial components are[2] (Supplementary Note 1)

$$\sigma_C^A = -\alpha f \text{ and } \sigma_C^R = -\tfrac{1}{2}\alpha f, \qquad (4)$$

respectively. They prompt the surface-induced elastic strain $\epsilon_C$, with axial and radial components[31]

$$\epsilon_C^A = \frac{\nu - 1}{Y}\alpha f \text{ and } \epsilon_C^R = \frac{3\nu - 1}{2Y}\alpha f. \qquad (5)$$

$Y$ and $\nu$ represent Young's modulus and Poisson's ratio, respectively, of the bulk. Consequences of this surface-induced elastic relaxation are firstly, a reduction in the energy of the surface regions by $f\delta e$, where $\delta e = \epsilon_C^A + \epsilon_C^R$ and secondly, an increase of the elastic strain energy density in the bulk by $\delta\Psi = \tfrac{1}{2}\epsilon_C : \sigma_C$. It is well known[32,33] that the energy increase in the bulk can only partly compensate the energy reduction at the surface. Inserting the two energy terms into Eq. (2), one indeed finds a reduced $\gamma$ of the relaxed nanowire,

$$\gamma_{\text{relaxed}} = \gamma_0 - \frac{3 - 5\nu}{4Y}f^2\alpha, \qquad (6)$$

where $\gamma_0$ refers to the unstrained surface.

**Energy balance and flow stress.** The contribution of capillarity to the flow stress of a nanowire, loaded axially, is readily obtained by evaluating Eq. (3) while using Eq. (6) for $\gamma$ and noting that $\delta\alpha/\delta\epsilon^p = \alpha/2$ for a long cylinder (elongation at constant $V$ increases the surface area). One thus obtains

$$\Delta T = \frac{1}{2}\alpha\left(\gamma_0 - \frac{3 - 5\nu}{2Y}f^2\alpha\right). \qquad (7)$$

The relaxation terms in Eqs. (6) and (7) are small in any case and negligible in experimental situations even for very small structures (Supplementary Note 2), suggesting that surface stress does not contribute significantly to $\Delta T$. Ignoring the $f$-dependent term in Eq. (7), we find that $\Delta T_S$, the change in flow stress due to the surface excess free energy, is simply

$$\Delta T_S = \frac{\gamma_0}{r}. \qquad (8)$$

This is the well-known relation behind zero-creep experiments. In view of Eq. (1) and of the sign convention, positive and negative stress in tension and compression, respectively, Eq. (8) suggests strengthening in tension yet weakening in compression, in other words, a tension–compression asymmetry of the contribution of the surface to stresses in small-scale plasticity.

Note that Eq. (8) accounts for an energy balance during plastic deformation and is not inherently related to the acting stresses in the nanowire. Those stresses are given by Eqs. (4), and they scale with $f$ rather than $\gamma$.

**Impact of surface stress.** Even though the energy-based considerations marginalize the role of surface stress, let us inspect a conceivable direct impact of that parameter on the deformation of a nanowire. Of relevance for plasticity is only the deviatoric part of $\sigma_C$. Equations (4) imply this to be a uniaxial stress, of magnitude $-\alpha f/2$, along the wire axis. This stress adds to that caused by the external traction. The Peach–Köhler forces on dislocations in the interior of the wire then see an extra contribution, analogous to the addition of an external stress $\Delta T_C$, which obeys

$$\Delta T_C = \frac{f}{r}. \qquad (9)$$

As explained in the Introduction, the reasoning behind Eq. (9) has given rise to suggestions—partly by one of the present authors—that the surface stress enhances the tensile strength of nanowires or may even prompt spontaneous plastic contraction[4–9,11–13]. Yet, this argument is problematic as it singles out the action of the bulk stress on the dislocations, thereby ignoring the action of the stresses near the surface.

Figure 1a and b illustrates how the continuum theory of capillarity decomposes the position-dependent net stress, $\mathbf{S}^{\text{net}}$, in a nanowire into the bulk stress $\sigma_C$ that acts throughout the cross-section and surface stresses that act along its perimeter. Let us here ignore this decomposition and relate the energetics of dislocation plasticity to the more fundamental quantity $\mathbf{S}^{\text{net}}$. In the absence of an external load, mechanical equilibrium requires that the area-integral of the traction, $\mathbf{t} = \mathbf{S}^{\text{net}} \cdot \mathbf{n}$, on a cross-section (unit normal $\mathbf{n}$) through the nanowire must vanish. One can readily confirm that the net mechanical work, which is done by the Peach–Köhler forces when a dislocation glides over the entire cross-section, scales with the integrated traction force (Supplementary Note 3) and so must vanish when there is no external load. Figure 1c illustrates the opposite-signed Peach–Köhler forces on dislocation segments in the bulk and

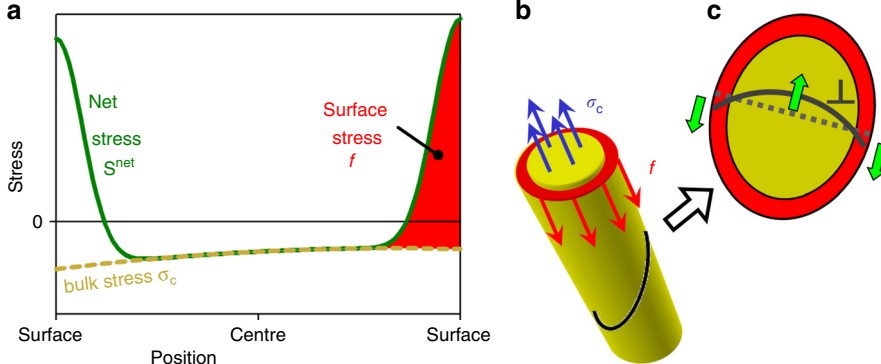

**Fig. 1** Stress in a nanowire and its impact on dislocation glide. Schematic representations. **a** Green line, stress profile along a linear section through the wire. The actual stress $\mathbf{S}^{\text{net}}$ may be decomposed into bulk stress $\sigma_C$ (yellow dotted line) and surface stress $f$ (red shaded area). **b** Balance of force on a normal cross-section. Surface regions (red) experience tensile stress, which is represented by the surface stress and which is compensated by an oppositely-signed surface-induced stress $\sigma_C$ in the bulk (blue). **c** Dislocation (gray line) on an inclined cross-section. Shear components of stresses from **a** give rise to Peach–Köhler forces that mutually compensate

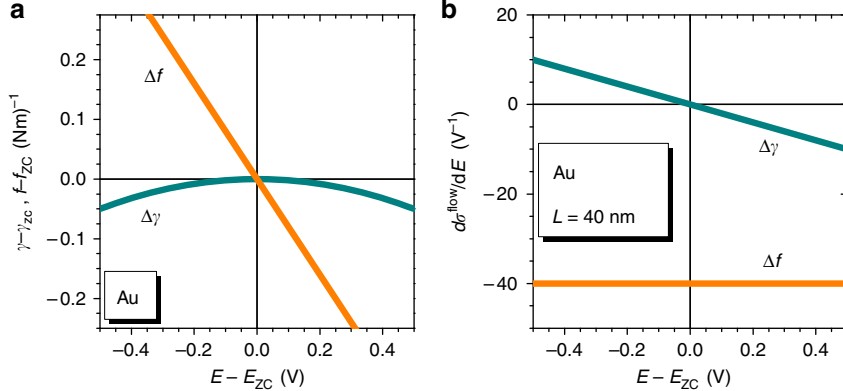

**Fig. 2** Capillary forces at gold surfaces and their impact on the flow stress of a nanowire. **a** Linear variation of surface stress, $f$, with electrode potential, $E$, around the potential of zero charge, $E_{zc}$, is distinguished from parabolic variation of the surface tension, $\gamma$. **b** Flow-stress potential coupling parameter, $d\sigma^{flow}/dE$, vs. $E$. Qualitatively different predictions for the coupling are obtained depending on whether $\sigma^{flow}$ is assumed to respond to changes in $\gamma$, Eq. (12), or in $f$. Graphs show extrapolated behavior based on the quadratic and linear approximations of Eqs. (10) and (11) along with experimental values for capacitance and electrocapillary coupling of gold near $E_{zc}$, see main text. **b** Assumes ligament diameter 40 nm, representative of the experiment

near the surface, as implied by the opposite-signed stresses in the respective regions. These forces act analogously on full dislocations and on partial dislocations that propagate a stacking fault or a twin. The lattice instability of small nanowires by twinning shears the entire cross-section by a partial dislocation Burgers vector. The work against the acting stresses is again governed by the area-integral of the traction[34], which vanishes.

A quite different result and specifically a nonvanishing work of deformation would be obtained if, erroneously, the net mechanical work was derived from the bulk stress $\sigma_C$ alone, excluding the stress in the surface regions from the consideration. This, however is the argument that leads to the prediction of a net contribution of the surface-induced bulk stress to dislocation plasticity, Eq. (9), and to the suggestion of a spontaneous plastic shear driven by surface stress. Clearly, that approach is not appropriate, and claims of a surface stress-induced strengthening or weakening of nanowires must be considered with caution or even rejected outright.

While dislocations from a stable or increasing population, for instance sustained from single-arm sources[35], may carry the plasticity and control the strength of small structures including NPG[8,36–38], nanowires may be dislocation-starved and their strength controlled by dislocation nucleation[6,39]. As nucleation is favoured at free surfaces of bulk materials[34] and nanowires[6,39], the nucleation events do not probe the surface-induced bulk stress that leads to Eq. (9) but they are at least partly affected by the large and opposite-signed stresses in the surface regions. This again sheds doubt on predictions, such as Eq. (9), for strengthening or weakening by surface stress, emphasizing the need for experiment.

**Discriminating between surface tension and surface stress.** Discriminating by experiment between the impact of surface stress and surface tension on the plastic flow of nanostructures is challenging as Eqs. (8) and (9) predict typically quite similar size effects. However, one may exploit that $\gamma$ and $f$ respond differently to changes in the electrode potential, $E$, or in its conjugate parameter, the superficial electric charge density (charge per area) $q$, if the surface is wetted by a fluid electrolyte. For a recent review of this electrocapillary coupling see ref. [30]

The Lippmann equation requires that $d\gamma = -qdE$. In as much as the capacitance, $c$, can be approximated as constant near the potential of zero charge, $E_{zc}$ (where $q = 0$), $\gamma$ varies

parabolically as

$$\gamma = \gamma_{zc} - \frac{1}{2}c(E - E_{zc})^2. \tag{10}$$

Contrary to $\gamma$, the surface stress may vary linearly near $E_{zc}$, so that (see Eq. (5.34) in ref. [30])

$$f = f_{zc} + \varsigma c(E - E_{zc}), \tag{11}$$

with $\varsigma$ the electrocapillary coupling coefficient. The values of $c$ and $\varsigma$ are well-established for gold surfaces in weekly adsorbing aqueous electrolytes, such as those of the present work, near $E_{zc}$. Here $c \approx 40\,\mu\text{F cm}^{-2}$ (ref. [40]); furthermore, $\varsigma$ is invariably negative-valued at transition metal surfaces near $E_{zc}$, and specifically $\varsigma = -2\,\text{V}$ for gold[3,30,41,42]. Figure 2a compares the variation of $\gamma$ and of $f$ for gold surfaces near $E_{zc}$.

The experiments in this work explore the variation of the flow stress, $\sigma^{flow}$, with $E$. Figure 2b summarizes the implications of our discussion for the coupling $d\sigma^{flow}/dE$, accounting for the numerical values of $c$ and of $\varsigma$ of gold near $E_{zc}$. If the surfaces affect the strength via surface stress, then Eqs. (9) and 11 imply $d\sigma^{flow}/dE = c\varsigma/r$. As $c > 0$ and $\varsigma < 0$ on clean transition metal surfaces, it follows that $d\sigma^{flow}/dE$ is negative throughout the potential regime of capacitive charging. By contrast, if surface tension is the relevant capillary parameter (Eq. (8)), then Eq. (10) implies that

$$\frac{d\sigma^{flow}}{dE} = -\frac{c}{r}(E - E_{zc}). \tag{12}$$

Here, the stress-potential coupling is positive at potentials negative of $E_{zc}$, yet the sign is inverted when $E_{zc}$ is crossed. The distinctly different predictions will allow us to discriminate, by means of in situ deformation experiments in electrolyte, between the two scenarios.

**In situ compression experiments.** As detailed in the "Methods" section, we prepared macroscopic samples of NPG with different mean ligament diameters, $L$, and with solid fractions $\varphi \sim 0.3$ by electrochemical dealloying. The setup of Fig. 3 allowed uniaxial compression tests in situ in electrolyte and under control of the electrode potential, $E$. Motivated by the distinctly different behavior of surface stress and surface tension during capacitive charging, see Fig. 2, we focused on potentials in the vicinity of $E_{zc}$. Our electrolytes, 0.7 M NaF, 1 M HClO$_4$, and 0.5 M H$_2$SO$_4$,

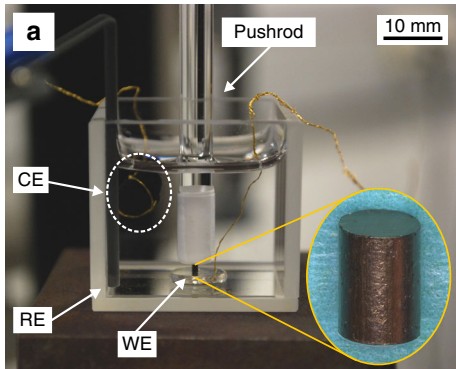
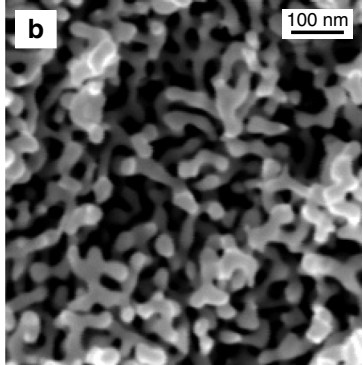

**Fig. 3** In situ setup for compression tests under potential control. **a** Nanoporous gold (NPG) sample (inset) forms the working electrode (WE) and is loaded by a glass rod. CE and RE: counter and reference electrodes. **b** Scanning electron micrograph of the NPG microstructure

comprise anions that adsorb nonspecifically on Au. Yet, the strengths of the gold–anion interactions differ[3]. Potential steps were imposed during compression. Cyclic voltammograms (Supplementary Note 4; Supplementary Fig. 1) show the region of dominantly capacitive charging to extend up to $E \leq 1.0$ V. Electrosorption of $OH^-$, involving up to one molecular monolayer[43], dominates at more positive $E$. All electrode potentials in this work are referred to the standard hydrogen electrode (SHE).

Figure 4 summarizes exemplary results of an in situ compression test, here for $L = 40$ nm and in 0.5 M $H_2SO_4$. The potential (blue line in Fig. 4a) was kept at 1.0 V up to 20% engineering strain, establishing a reference for deformation at constant potential. $E$ was then stepped to its most negative value, 0 V, and a series of potential holds, separated by 100 mV, followed. The step sequence was inverted when $E$ reached 1.5 V. The stress–strain graph in Fig. 4a illustrates how the experimental (flow-) stress, $\sigma$, reacts to the variation of $E$. The large deformability and pronounced strain hardening agrees with findings for compression of NPG in air[22]. The most obvious consequence of the potential variation is the strong change in $\sigma$ during oxygen electrosorption (shaded regions in Fig. 4), well compatible with the observations in ref. [27] Yet, the focus of the present study is on the capacitive processes. The jumps in stress are here smaller, but the enlarged graphs of Fig. 4b and c shows that they are well detectable.

Remarkably, the flow stress magnitude is diminished during the jumps at low potential (Fig. 4b) but it is enhanced at higher potential (Fig. 4c). This behavior is more obvious when inspecting the flow-stress potential response parameter, $\delta\sigma/\delta E$, in Fig. 4d. In the capacitive regime $\delta\sigma/\delta E$ starts out positive-valued at negative $E$. Increasing $E$ lets the response approach zero and then change sign at around 0.7 V.

The sign change of $\delta\sigma/\delta E$ is remarkable in view of the inversion of the potential-response of the surface tension at the potential of zero charge, $E_{zc}$, see Eq. (10). The value of $E_{zc}$ is characteristic of the combination of electrolyte and of the electrode surface's crystallography and defect structure. Using two independent variants of the immersion technique (see "Methods"), $E_{zc}$ of NPG in 0.5 M $H_2SO_4$ was determined as $0.70 \pm 0.14$ V. This value indeed coincides with the potential where $\delta\sigma/\delta E$ inverts its sign, see Fig. 4d.

The bold solid line in Fig. 4d represents the prediction of Eq. (12) for $\delta\sigma^{flow}/\delta E$ of gold nanowires with $L = 40$ nm, accepting $\gamma$ and not $f$ as the governing capillary parameter and using $c = 40$ $\mu F\,cm^{-2}$ (ref. [40]). It is striking that, around $E_{zc}$, the slope of the experimental graph is in excellent agreement with the prediction.

**Varying the ligament size.** Figure 5 summarizes the results of in situ compression tests with different $L$. Strength and flow stress

increase with decreasing $L$, in agreement with previous reports[22,24–26]. We again focus on $\delta\sigma/\delta E$ during capacitive charging, see Fig. 5b. The general trends agree well for all $L$, yet the response is stronger for smaller $L$. The size-dependence is anticipated by Eq. (12), and indeed the solid lines - which represent that equation in Fig. 5b (no free parameters) - agree quantitatively with the experiment.

Figure 5b also shows that $\delta\sigma/\delta E$ exhibits the same size-dependent trend as $\sigma$. To verify this observation, Fig. 5c plots the response parameter normalized to the flow stress, $\sigma^{-1}\delta\sigma/\delta E$, vs. $E$. The graphs nearly coincide in this representation, irrespective of $L$. Apparently, there is a link between phenomena responsible for strengthening by electric potentials and those governing the size-dependence of the strength.

**Varying the anion.** Besides $H_2SO_4$, we also explored two other electrolytes, aqueous 1 M $HClO_4$ and 0.7 M NaF. Figure 6a shows the corresponding normalized response parameters $\sigma^{-1}\delta/\delta E$, along with the result for $H_2SO_4$. The ligament size was 40 nm for each sample. To remove the impact of the different $E_{zc}$ in the individual electrolytes, all electrode potentials were referred to the respective $E_{zc}$. The $E_{zc}$ for our studies with $SO_4^{2-}$, $ClO_4^-$, and $F^-$, as determined by the immersion method (see Fig. 6b and "Methods") were $0.70 \pm 0.14$, $0.77 \pm 0.19$, and $1.02 \pm 0.22$ V. Figure 6a shows that highly reproducible results, independent of the anion, are obtained when $\sigma^{-1}\delta\sigma/\delta E$ is plotted vs. $E - E_{zc}$. Specifically, the trend for the response to change sign at $E_{zc}$ appears generic.

**Small strain and inverse scan direction.** Results of additional in situ compression tests (Supplementary Note 5; Supplementary Fig. 2) confirm that the sign-inversion of the flow-stress potential-response is recovered when scanning twice through $E_{zc}$ and that at all strains (down to values as small as 4%) the sign of the response is consistent with Eq. (12) and with the prediction of the $\Delta\gamma$ graph of Fig. 2. Thus, all experiments support surface tension as the relevant capillary force. The stronger and oppositely-signed response that would indicate surface stress as relevant (see the $\Delta f$-graph in Fig. 2) is not supported by the experiment.

**Evolution of surface area during deformation.** Our theory presupposes that plastic deformation changes the net surface area, $A$. To verify this notion, we used the electrochemical capacitance ratio method (Supplementary Note 6) to monitor the evolution of $A$ during compression experiments such as Figs. 4 and 5. The results, as described by Supplementary Fig. 3a, confirm that $A$ diminishes continuously during compression. The relative

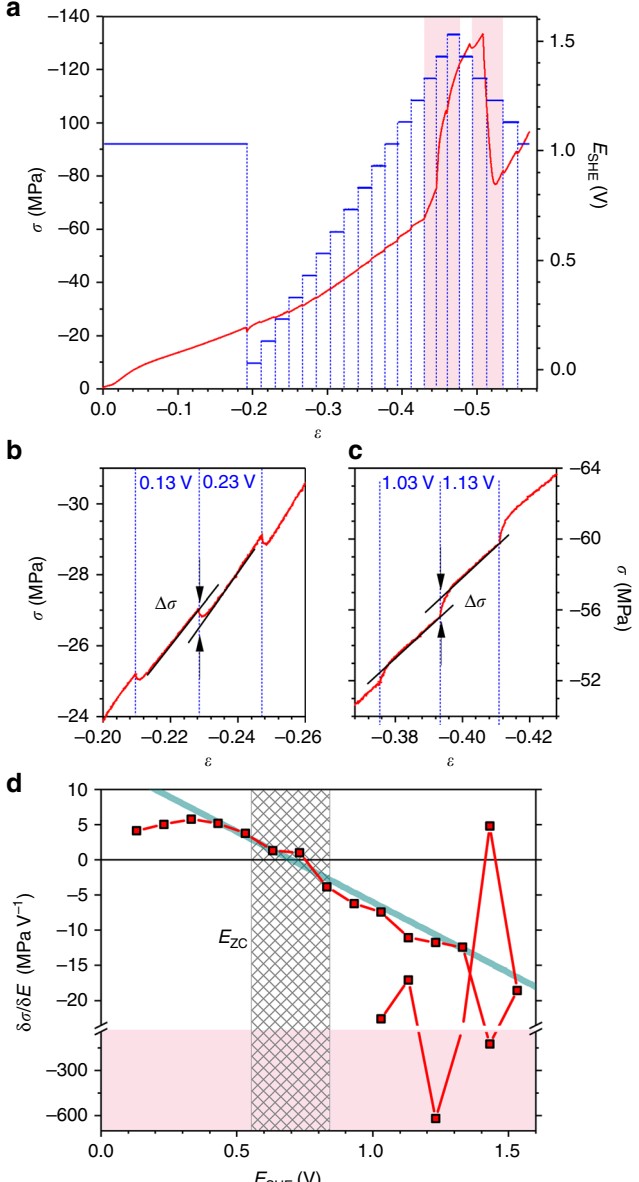

**Fig. 4** Results of an in situ compression test. **a** Red, stress $\sigma$ vs. strain $\varepsilon$ at constant engineering strain rate $10^{-5}\,s^{-1}$. Potential, $E$, vs. SHE (*blue*) is superimposed. **b**, **c** Details from **a**, showing increase or decrease of $\sigma$ in response to potential steps during capacitive charging. **d** Response, parameterized as $\delta\sigma/\delta E$, of experimental flow stress to jumps in $E$. Note axis break and different scales for the capacitive-regions and OH-regions. Blue solid line: predicted response, Eq. (12); note excellent agreement near potential of zero charge, $E_{zc}$. Hatched: experimental range of $E_{zc}$. Red shaded regions in **a**, **d** denote regimes of OH-adsorption/desorption. Test in 0.5 M $H_2SO_4$, mean ligament diameter 40 nm

change in $A$ (Supplementary Fig. 3b) during compression is in fact consistent with the atomistic simulation of ref. [8] Electron micrographs (Supplementary Note 6; Supplementary Fig. 4) rule out potential-induced coarsening as an origin of the variation. Thus, besides densifying the ligament network[8,36], the plastic compression changes the microstructure by reducing the ligament aspect ratio, which decreases the net surface area. Furthermore, previous experiment[36,37] and atomistic[8] as well as continuum simulation[38] suggest that compression also enhances the dislocation density.

## Discussion

Our discussion starts out with apposing experiment and theory. Contrary to suggestions in previous work, our theory finds no forceful argument for a significant impact of surface stress on the plastic flow of nanowires. Our arguments rest on firstly, an explicit consideration of the local stress state of the material near the surface and of its impact on dislocation plasticity and secondly, the analysis of the energy of the deformed nanowire, in which the contributions of the surface tension dominate while contributions due to surface stress-induced relaxation are negligible. In as much as surfaces contribute a driving force for plastic deformation, the more obvious key parameter is the surface tension $\gamma$ and not the surface stress $f$.

By deforming NPG wetted by electrolyte under potential control, our experiments probe the flow stress variation while $\gamma$ and $f$ can be varied simultaneously yet in distinctly different manner. NPG, when unloaded at any state of plastic flow and then reloaded, yields at the last flow stress before the unload (see ref. [22], Supplementary Note 7, and Supplementary Fig. 5). Thus, the flow stress at any state of plastic strain agrees with the yield stress in that state; this connects our experimental investigations of plastic flow to the strength of nanostructures.

The experiment supports our theory: First, for negatively charged surfaces, our experiment finds a positive-valued coupling, $d\sigma^{flow}/dE$, between flow stress and electrode potential. The sign agrees with the prediction based on $\gamma$ as the relevant capillary parameter but is incompatible with the potential-dependence of $f$ (see Fig. 2). Second, the sign-inversion of $d\sigma^{flow}/dE$ at the potential of zero charge, $E_{zc}$, in our experiment is consistent with the variation of $\gamma$ as embodied in the Lippmann equation and it disagrees with the expectation that $d\sigma^{flow}/dE$ should be negative-signed at all potentials if $f$ were the controlling quantity. Third, our in situ capacitance data show the surface area to decrease during plastic deformation, in agreement with the premises behind the analysis of $\gamma$ as a driving force for plastic compression. The decrease of the surface area of NPG with strain is least pronounced in the early stages of deformation, see ref. [8] and Supplementary Fig. 3. Consistent with our theory, the flow-stress potential-response also tends to be less pronounced at small strain (see the data at most negative potential in Figs. 5c and 6a, and Supplementary Fig. 2d). Fourth, as the most compelling evidence, the potential-dependence and size-dependence of the experimental coupling strength near $E_{zc}$ agree quantitatively with the prediction of Eq. (12).

Determining potentials of zero charge is notoriously challenging and this motivates a critical inspection of our findings concerning this parameter. $E_{zc}$ of single- and polycrystalline gold surfaces in similar electrolytes tend to be 200 to 500 mV more negative than our data. For instance, capacitance measurements in 0.01 M $HClO_4$ suggest 470 and 320 mV for bulk-truncated Au (111) and Au(100), respectively[44]. Immersion measurements of $E_{zc}$ for polycrystalline gold in dilute $HClO_4$, $H_2SO_4$, or NaF solutions have been found even more negative, from 170 to 330 mV (ref. [45]). The more positive values of our study may suggest that the extremely high defect density (step edges, kinks) —which is required by the curvature of the surfaces of NPG–shifts $E_{zc}$ to positive. Our data is also qualitatively consistent with the expected compressive strain (Eq. (5)) in NPG and with the negative-valued $\varsigma$. Furthermore, the order of the $E_{zc}$ for the different anions $(SO_4^{2-} \lesssim ClO_4^- < F^-)$ agrees well with literature data[46–48]. Lastly, in agreement with theory, the results for $d\sigma^{flow}/dE$ in experiments with the individual ion species coincide precisely when plotted vs. $E - E_{zc}$, see Fig. 6a. Thus, even when the uncertainties involved in determining $E_{zc}$ for real surfaces of the nanoporous metals in our compression experiments are

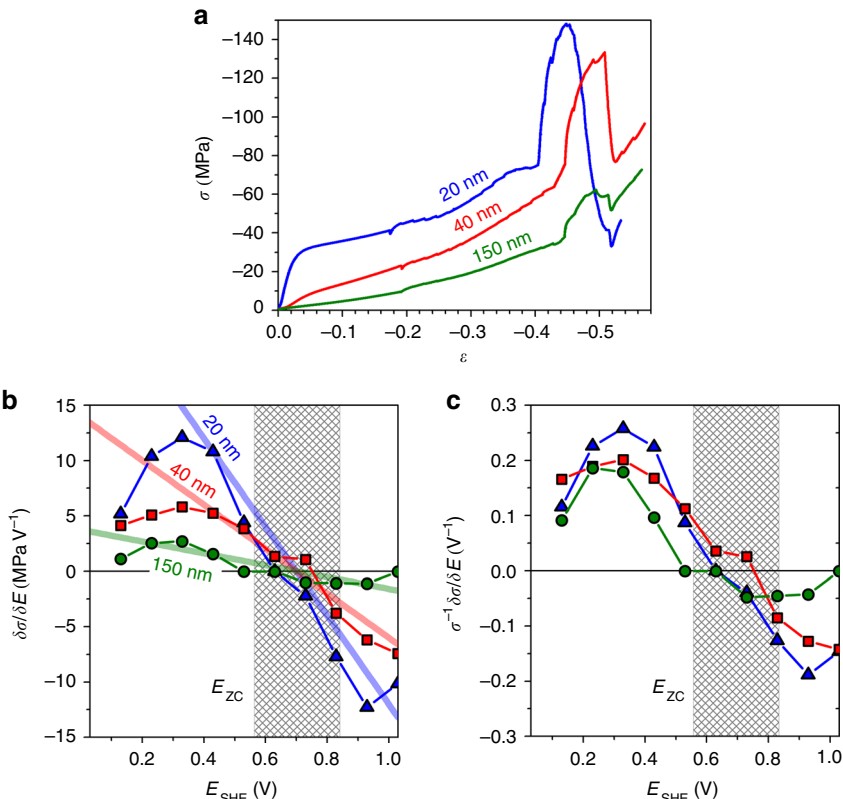

**Fig. 5** Experiments with different ligament size. **a** Flow stress $\sigma$ vs. strain $\varepsilon$ during deformation with strain rate $10^{-5}$ s$^{-1}$ in compression. **b** Effective response of $\sigma$ to jumps in the electrode potential, $E$, determined as $\delta\sigma/\delta E$. Straight lines: prediction by Eq. (12). **c** The parameter $\delta\sigma/\delta E$ is normalized to the actual value of flow stress, $\sigma_0$. Note no size-dependent behavior in this case. Shaded: range of potential of zero charge, $E_{zc}$. Test in 0.5 M H$_2$SO$_4$; ligament sizes are indicated by labels

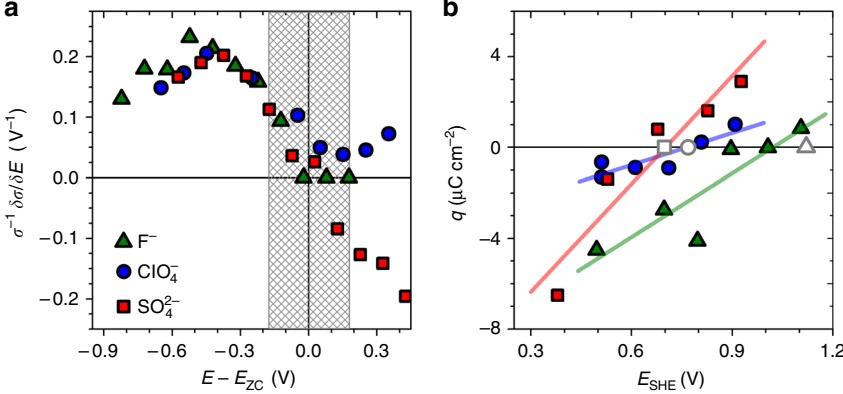

**Fig. 6** Experiments with different anions. **a** Normalized flow-stress electrode-potential coupling parameter $\sigma^{-1}\delta/\delta E$, plotted vs. difference, $E - E_{zc}$, between electrode potential and the potential of zero charge, $E_{zc}$. Note the excellent agreement in the capacitive regime. **b** Immersion charge density, $q$, vs. electrode potential. Intersects of linear fits with the abscissa provide $E_{zc}$. Open symbols: $E_{zc}$ from separate experiments using open-circuit immersion, see "Methods". Shaded: error range for $E_{zc}$. Mean ligament diameters are 40 nm; anions are indicated in legend

acknowledged, the general magnitude of our $E_{zc}$ and the variation with the anion species appear robust.

The above arguments support the notion that the minimum of the flow stress magnitude (at constant strain rate) in our compression experiments is connected to $E_{zc}$. This is compatible with early experiments on macroscopic metal wires, which exposed a minimum of the tensile creep rate (at constant stress) at $E_{zc}$ (refs. [19,49]). Both findings are indeed consistent with the theory, if the predicted tension–compression asymmetry is born in mind.

We now appose conservative to dissipative processes of deformation. The analysis of conservative (energy-related) processes during plastic flow leads to the prediction for the change in flow stress due to the surface excess free energy (or surface tension, $\gamma$), Eq. (12). The predicted change is positive, suggesting a tension–compression asymmetry with strengthening in tension and weakening in compression. The well-established finding that smaller is stronger in both, tension and compression, implies that additional, dissipative strengthening processes act in a symmetric manner. The nature of the dissipative processes is not the subject of our work. However, it is significant that we find the potential-response of the flow stress to scale with the flow stress itself, even for different structure size where values of $\sigma^{\text{flow}}$

differ significantly because smaller is stronger. This scaling at least rules in that the dissipative contributions to $\sigma^{flow}$ are also related to the surface, a scenario that would result in a size-dependent $\sigma^{flow}$. In fact, the scenario is compatible with previous observations from in situ tests in electrolyte studying the impact of the specific adsorption of $OH^-$ ions. These experiments where rationalized[27] in terms of a dissipative adsorption locking mechanism[50] where adsorbate impedes the motion of the dislocation end points.

Slip traces and surface roughness are relevant for our discussion. With an eye on capacitive processes, as in the present experiment, it has been pointed out that dislocation end points moving along the surface of an idealized crystal with planar facets create slip traces that increase the surface area. Reducing $\gamma$ by capacitive charging would thus reduce an energy barrier for plastic deformation, enhancing the deformation rate or reducing the flow stress magnitude[19,27,51]. The argument agrees with our theory inasmuch as mechanical work is again balanced against the work required to increase the surface area. Yet, the conclusions differ: the reduction of $|\sigma^{flow}|$ upon charging is here predicted irrespective of whether the plastic strain is in compression or in tension.

The in situ compression tests on NPG do not support the slip-trace argument: $|\sigma^{flow}|$ increases upon charging and deformation decreases the surface area. The disagreement was noted by Jin and Weissmüller[27] based on first tentative experiments into the potential-dependent flow behavior of NPG near $E_{zc}$. On that basis, the authors rejected arguments balancing mechanical work against surface tension as apparently not relevant for the potential-dependent flow. Yet, the apparent contradiction is naturally resolved when one realizes that real surfaces are typically rough and exhibit many pre-existing step edges and kink sites. Dislocations moving along the surface may then not only create new step edges—as they invariably do on planar terraces—but also remove pre-existing ones[15]. The continuum picture of our theory here appears appropriate; it describes the geometry through the radius $r$ and, thereby, through surface mean curvature. Curvature requires edges, hence roughness, thereby connecting to the atomic-scale picture in a statistical sense. Elongation at constant volume increases the curvature and hence the roughness as well as the net surface area, whereas compression has the opposite effect. These considerations imply that experimental investigations of flow-stress potential coupling for nanostructures with faceted surfaces—as opposed to the rough ones of the present material—might reveal qualitatively different behavior. Surfaces of NPG which were originally rough have been observed to reconstruct and to develop microfacets when the material acts as a catalyst for CO oxidation[52]. In situ mechanical tests under controlled gas atmosphere might thus probe a possible distinction between the strength of nanostructures with rough or faceted surfaces in future studies.

As outlined in the Introduction, extremely small nanowires or the very small ligaments of some NPG studies can experience spontaneous irreversible contraction even when there is no external load. Plastic yielding prompted by surface stress has been invoked to explain the observation. Yet, our study does not support surface stress-induced yielding. Instead, since the energies of the initial and final states are governed by the surface tension, it appears appropriate to identify surface tension as the driving force. This in itself does not explain the microscopic mechanism of the spontaneous yielding, since the stresses in the solid are not governed by $\gamma$. Further studies of the issue would seem to be of high interest.

In summary, our experiment provides compelling support of the theory suggesting substantial effects of surface tension on plastic flow at the nanoscale, while rejecting significant contributions by surface stress. This suggests that the impact of capillarity on the size-dependent yield strength and the tension–compression asymmetry of small structures might need to be reconsidered.

## Methods

**Preparation of NPG by dealloying.** Master alloys $Au_{25}Ag_{75}$ were prepared according to ref. [27], except that wire drawing and sectioning by a wire saw were used to make cylindrical samples, 1.17–1.37 mm in diameter and 1.90–2.10 mm in length. Electrochemical dealloying in 1 M $HClO_4$ (60% $HClO_4$, ACS grade, Merck) at ambient temperature used a potential of 1.26 V vs. SHE. Reference and counter electrodes (RE and CE) for dealloying were pseudo Ag/AgCl (+0.515 vs. SHE, "HydroFlex", Gaskatel) and a coiled Ag wire (99.9985%, Alfa Aesar), respectively. Using 1 M $HClO_4$ prepared from high purity $HClO_4$ (70%, Suprapur, Merck), the as-dealloyed samples were reduced during 15 potential cycles between 0.01 and 1.01 V with a scan rate of 5 mV s$^{-1}$; this lead to mean ligament diameter $L\sim20$ nm. Cycling 15 times between 0.01 and 1.51 V at 5 mV s$^{-1}$ led to $L \sim 35$–45 nm. Rinsing with ultrapure water (Ultra Clear TWF UV TM, Siemens) and drying for >2 days in Ar (5.0) flow at room temperature followed. Samples with $L \sim 150$ nm were prepared by annealing $L = 20$ nm NPG at 400 °C for 1 h in Ar. Solid fractions, as determined from external sample dimensions and mass, were $\varphi = 0.27 \pm 0.01$, $0.29 \pm 0.02$, $0.32 \pm 0.03$ for samples with $L = 20, 40, 150$ nm, respectively. Mean ligament diameters, $L$, were estimated from scanning electron micrographs. All potentials in this work are referred to the standard hydrogen electrode (SHE).

**Immersion method.** As an approach to the potential of zero charge, $E_{zc}$, of NPG under experimental conditions, we exploited the large surface area and used two variants of the immersion method[53,54]. Electrochemically reduced and dried NPG with $L = 40$ nm was connected as the WE of a three-electrode cell, initially without contact to the electrolyte, and was then rapidly immersed (for wetting kinetics see ref. [55]). In the open-circuit immersion method, zero-current galvanostatic control was maintained, and the initial value of the potential transient supplied $E_{zc}$. The charge integration method studies immersion under potentiostatic control at a series of fixed immersion potentials, $E_{im}$. $E_{zc}$ was determined by linear regression using the net charges, $Q(E_{im})$, as determined by integration of the current transient. Figure 6b shows that the two methods are highly consistent, for example, the charge integration suggests $E_{zc} = 0.77 \pm 0.19$ V (errors derived from linear regression) for $HClO_4$, as compared to $0.77 \pm 0.10$ V (errors derived from scatter for repeated experiments) from open-circuit immersion.

**In situ mechanical testing.** Compression tests at ambient temperature used a Zwick Z010TN frame, at constant engineering strain rate 10$^{-5}$ s$^{-1}$. The strain was measured by a laser extensometer. The in situ setup (Fig. 3) used a glass cuvette filled with electrolyte with a cold-worked Pt plate as lower load surface and electrical contact. The load was applied via a glass rod. A gold wire wrapped with porous carbon cloth formed the CE, Ag/AgCl (Dri-Ref, World Precision Instruments) served as the RE, and a potentiostat (PGSTAT 302N, Metrohm) controlled the potential. Throughout this work, we specify engineering strains and stresses. As cross-sections vary little[36,56], engineering and true stresses agree closely.

**Data availability.** The relevant data are available within the article and its Supplementary Information file or from the authors on reasonable request.

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

## Acknowledgements

This work was supported by Deutsche Forschungsgemeinschaft through SFB 986 "Taylor-Made Multiscale Materials Systems—M$^3$", subprojects B2 and B8.

## Author contributions

N.M. designed and performed the experiments, analyzed the results, and wrote the manuscript. J.M. designed the testing protocols and J.W. designed the theory. All authors discussed results and manuscript.

## Additional information

**Competing interests:** The authors declare no competing financial interests.

