## [Peer Review File · Nature Communications]

Reviewers' comments:

Reviewer #2 (Remarks to the Author):

I admit to having reviewed this manuscript earlier for **** and still have a few of the same problems with the current form of the manuscript as I had before. Consequently, the authors have seen some of my comments below and have not addressed them in this current version of the manuscript.

In the Introductory paragraph, the authors of this manuscript discuss how surface stress, f , induces a large bulk stresses that can affect shear processes in nanoscale solids. They correctly argue that such effects have been seen in computer simulation, but also erroneously state that such effects have also been observed in experiment [their citations 9&10]. I have re-examined their cites 9 & 10 and can not find where they claim to have "observed spontaneous plastic shear". They merely argue that this is a possibility based on seeing a higher density of dislocations for smaller scale NPG than observed in larger scale NPG structures. Another possible explanation for this is that gold leaf is produced by hammering and so has a huge dislocation density. Small-scale NPG structures forming by dealloying of the leaf (at high potentials and consequently high rates) simply allows for the preservation of more dislocations than the larger scale NPG structures (formed at lower potential and lower rates). To my knowledge the authors of cite 9 did not characterize the starting dislocation structure of un-dealloyed gold leaf.

They also say in the Introduction that "here we critically examine these notions", but actually they don't since the fundamental issues discussed in the Introduction (tension/compression asymmetry and spontaneous yielding) are not necessarily connected to the issue actually investigated in the current manuscript (i.e., the identification of which surface parameter (f or γ) affects the flow stress of a nanomaterial). This is clear from the authors' own commentary in the concluding remarks of the manuscript.

"This in itself does not explain the microscopic mechanism of the spontaneous yielding, since the stresses in the solid are not governed by γ . Further studies of the issue would seem to be of high interest."

Finally, with regard to the commentary in the Introduction and Conclusions, I'm sure that the authors would agree that the surface stress effects they consider (e.g., with regard to tension compression asymmetry, etc.,) are really not expected to be significant for length scales above ~ 5 nm.

Having said all this, in my view the authors' interpretation of the experimental results shown in Figure 4 of the manuscript is correct and connected to the effect of potential (and adsorption) through changes in the excess surface free energy. Their Eq. (10) is important for understanding the general form of the experimental result. However, they could and should provide a much simpler and straightforward derivation of this equation.

In summary, I believe that the authors do arrive at the correct interpretation of their experiments. However, as described above, I have some real problems with some commentary in the Introduction of the manuscript that I find to be a bit misleading. If the authors address these concerns I would be able to recommend publication of the manuscript in Nature Communications.

Reviewer #3 (Remarks to the Author):

The authors present a study that focusses on the effect of surface stress and surface tension on the nano-mechanical behavior of crystals. It is argued that the current belief in this field of research is that capillary forces may affect the mechanical behavior at the nano-scale. This may lead to spontaneous shear. Due to the particular surface stress state, a tension-compression asymmetry is believed to emerge. The manuscript concludes that it is not the surface stress, but the surface energy, measured via the surface tension, that is the controlling parameter. The manuscript is well written and structured, but appears rather as a PRB than a body of work that would warrant publication in a journal for the broad general audience. In addition, it seems that some proposed ideas for the origins of the specific mechanical response of nano-porous gold (npAU) have been postulated by some of the authors and are now rejected. That means, the authors seem to contradict their earlier propositions. There is a priori nothing wrong with this, but NatComm does not seem to be the medium to discuss and address such matters. I would be more suitable in a specialized journal.

The theory part is elaborate, and does present the motivation for the experiments in a concise way. There are numerous steps that include quite some simplifications, which later seem to not matter, as the experiments are in agreement with the predictions of equation 12. For example, I find the picture of a dislocation gliding in the nanowire somewhat contrived. First of all, I am not aware of any studies that actually have revealed the presence of any mobile full dislocation segments at such small length scales. Secondly, dislocations in Au nanowires are mostly partials, where the nucleation stress is the rate limiting step for plasticity. As such, the discussion around the PK force and some dislocation mobility/traction force in the wires cannot be the relevant measure.

The experiments are quite innovative, and interesting. There are, however, several aspects that remain unclear in the manuscript. From figure 4a, I understand that the plastic strain is more than 50%. The regime in which equation 12 fits the data is for an E_{SHE} between 0.3 and 1.3 V. This is the strain regime of somewhere from 20 to 45%. This is a very high amount of plastic strain, and given the structure shown in figure 3b, I would expect that the npAu is compacted during compression. As the authors state, this would reduce the net surface area during the mechanical experiment. What is going on in the initial part of the stress-strain regime? There are ca. 20% of plastic strain that do not seem to be captured very well by the model. This is way beyond the elastic regime, and I do not see any plausible reason for the strong deviations between the prediction and the actual data. Actually, one would need to ask the question what makes the strain regime beyond 20% special to allow agreement with equation 12? Given the strong conclusions made by the authors, I am somewhat concerned about this fact.

In the discussion, the authors raise again the topic of a tension-compression asymmetry. Their experiments are done in compression, and not in tension. In order to justify the reoccurring focus on this matter, the authors should also include tension data. As it stands, there is no evidence or data that would allow clarifying the origin of this asymmetry. In general, the discussion is probably a bit long. After all, the theory part already covered many of the raised aspects, and the experimental section makes most statements clear.

Reply to the reviewers' comments

Reviewer #2 (Remarks to the Author):

I admit to having reviewed this manuscript earlier for ***** and still have a few of the same problems with the current form of the manuscript as I had before. Consequently, the authors have seen some of my comments below and have not addressed them in this current version of the manuscript.

In the Introductory paragraph, the authors of this manuscript discuss how surface stress, f , induces a large bulk stresses that can affect shear processes in nanoscale solids. They correctly argue that such effects have been seen in computer simulation, but also erroneously state that such effects have also been observed in experiment [their citations 9&10]. I have re-examined their cites 9 & 10 and can not find where they claim to have "observed spontaneous plastic shear". They merely argue that this is a possibility based on seeing a higher density of dislocations for smaller scale NPG than observed in larger scale NPG structures. Another possible explanation for this is that gold leaf is produced by hammering and so has a huge dislocation density. Small-scale NPG structures forming by dealloying of the leaf (at high potentials and consequently high rates) simply allows for the preservation of more dislocations than the larger scale NPG structures (formed at lower potential and lower rates). To my knowledge the authors of cite 9 did not characterize the starting dislocation structure of un-dealloyed gold leaf.

The relevant passages from references 9 and 10 are reproduced at the end of this reply. Their inspection shows that both experimental papers report irreversible macroscopic dimension changes without external load and propose them as evidence for spontaneous plastic deformation. They then speculate on surface stress as a possible origin. This, together with the stronger statements in the simulation studies (our references [4-8]), does motivate our statement that the state of the art embraces spontaneous plastic deformation at small structure size and that surface stress is proposed as the origin.

In response to the reviewer's comments, we have modified the text and now write

in the introduction: "Indeed, an instability to spontaneous irreversible contraction is observed in extremely small structures. Atomistic simulation studies [4-8] and, on a more speculative note, experimental reports [9,10] attribute the instability to plastic shear prompted by the action of the surface-induced stress."

in the Conclusions: "... extremely small nanowires or the very small ligaments of some NPG studies can experience spontaneous irreversible contraction even when there is no external load."

They also say in the Introduction that "here we critically examine these notions", but actually they don't since the fundamental issues discussed in the Introduction (tension/compression asymmetry and spontaneous yielding) are not necessarily connected to the issue actually investigated in the current manuscript (i.e., the identification of which surface parameter (f or γ) affects the flow stress of a nanomaterial).

What is the claim of our manuscript? Here is a longer version of the manuscript's passage cited by the reviewer: "Thus, surface stress is believed to impose a lower limit on the stable size of crystals and to contribute substantially to nanoscale mechanical behavior. Here, we critically examine those notions."

The claim is, investigate the suggested impact of surface stress on stability (spontaneous shear) and deformation behavior. This we do by means of a combination of theory and experimental investigations probing the impact of surface parameters on the flow stress. For more clarity, we have reformulated the above passage as follows:

"Thus, surface stress is believed to impose a lower limit on the stable size of crystals and to contribute substantially to nanoscale mechanical behavior. Here, we critically examine the suggested impact of the

surface stress.”

How are our observations directly linked to a tension-compression asymmetry? May we first emphasize that the link between the action of the capillary forces and the tension compression asymmetry is NOT an original point of our paper. As we point out quite clearly in the introduction, the asymmetry is firmly anchored in the existing literature, it is explicitly addressed in references 6-8 and 11-14. Together, these references alone have accumulated 488 citations in Web of Science, so this is relevant current opinion. The criticism of this point disregards the state of the art and it is poorly matched to our work.

The manuscript has been modified in order to more obviously communicate the link between the action of the capillary forces and tension-compression asymmetry. Besides adding an extra reference (reference 14), we now emphasize that the asymmetry is already part of the well-established observations on zero creep. This is explicitly stated in the last sentence of the following new passage, inserted in the third paragraph of the Introduction:

“The stress which is required for compensating the trend for contraction -- resulting in zero creep rate -- in a wire of radius r is tensile and of magnitude γ/r [15,16]. Zero creep experiments thus exemplify that tension-compression asymmetry results from the action of capillarity: creep is arrested by tensile stress but would be accelerated by a compressive stress of same magnitude.”

Why are our investigations of plastic flow linked to yielding? During plastic deformation, the microstructure of the material evolves and so do all its mechanical properties, including the yield stress. Mechanical tests that include unload/reload segments show that, at any state of plastic pre-strain, the yielding during reloading occurs precisely at that stress value which determines the flow stress before unloading. This observation establishes conclusively (for our material) that the flow stress at any given state of plastic strain represents the yield stress of the material in the respective state. The correlation between flow and yield is by no means exotic and we would be astonished if its use in our argument gave rise to debate with an expert and objective reviewer.

The revised manuscript contains an added figure (and added explanatory text) in the Supporting Online Material, figure which – as we trust – irrefutably establishes the correspondence of flow stress and yield stress in our material.

New Fig S5 in the Supporting Online Material:

Caption text:

Load-unload stress-strain data of dry NPG with $L = 40$ nm in compression. Engineering strain rate 10^{-4} s^{-1} . During reloading, the material yields at the stress value that marks the flow stress immediately prior to unloading.

Explanatory text was added in the SOM (not shown here for brevity).

This is clear from the authors’ own commentary in the concluding remarks of the manuscript. “This in itself does not explain the microscopic mechanism of the spontaneous yielding, since the stresses in the solid are not governed by γ . Further studies of the issue would seem to be of high interest.”

Even though our work does not answer all questions once and for all, why is it relevant? The very essence

of our paper is that it corrects the erroneous concept in the literature with respect to the driving force. We also consider it good practice to state that the atomic mechanisms have not been identified. In fact, atomic mechanisms may differ depending on details such as crystal structure, shape of the nano object and the resulting anisotropy of the stress tensor, stacking fault energy and so on. By contrast, the driving force argument is both forceful and general, applicable irrespective of atomic scale details.

Therefore, we maintain that our insights into driving forces conclusively clarify the most fundamental aspect of the problem, thereby providing a substantial new insight that qualitatively advances the science.

Finally, with regard to the commentary in the Introduction and Conclusions, I'm sure that the authors would agree that the surface stress effects they consider (e.g., with regard to tension compression asymmetry, etc.,) are really not expected to be significant for length scales above ~ 5 nm.

Up until the present experiments we would have entirely agreed. Yet, the results in our manuscript now suggest a different view: The effective stresses related to the surface diminish with increasing size as $1/r$, but so does the observed strength. In other words, the relative contribution of the surface effects is independent of the size in the entire range of our study, up to 150 nm. That finding is perfectly consistent with the observation (see our references [15,16]) that the impact of capillary forces on the mechanical behavior is measurable in metal wires with tens of micron in diameter, 10000 fold larger than the 5 nm mentioned by the reviewer. Irrespective of the above remark, structures at the 5 nm scale are of the most intense current interest in experiment and modeling, as documented by our references [4-14, 17].

In summary, we see our findings as highly relevant and we tend to qualify the reviewer's remarks as not impairing the significance of our paper.

Having said all this, in my view the authors' interpretation of the experimental results shown in Figure 4 of the manuscript is correct and connected to the effect of potential (and adsorption) through changes in the excess surface free energy. Their Eq. (10) is important for understanding the general form of the experimental result. However, they could and should provide a much simpler and straightforward derivation of this equation.

In order to provide the reader with a simpler approach to the subject, **we have added a new passage (which has already been shown above)** in the revised Introduction. The new passage provides a simple access to the effective extra stress that scales with γ/r (our Eq (8), the crucial intermediate result in the derivation) and to its implications.

Note that Reviewer #3, along with another reviewer from the earlier review round at Nature Materials, embraces our Theory section. Their comments also address certain details of the derivation as relevant. We would consider it inappropriate to hide these details by presenting only a simplified derivation. The level of sophistication in our Theory section is such that an educated reader from a general materials science/materials chemistry/solid-state physics community should readily be able to follow it. Claiming a new materials law without presenting substantiated underlying arguments from both, theory and experiment, is simply not up to the authors' scientific standards.

In summary, I believe that the authors do arrive at the correct interpretation of their experiments. However, as described above, I have some real problems with some commentary in the Introduction of the manuscript that I find to be a bit misleading. If the authors address these concerns I would be able to recommend publication of the manuscript in Nature Communications.

As the above comments emphasize, we believe to have modified manuscript so as to comply with all suggestions by this reviewer.

Reviewer #3 (Remarks to the Author):

The authors present a study that focusses on the effect of surface stress and surface tension on the nano-mechanical behavior of crystals. It is argued that the current belief in this field of research is that capillary forces may affect the mechanical behavior at the nano-scale. This may lead to spontaneous shear. Due to the particular surface stress state, a tension-compression asymmetry is believed to emerge. The manuscript concludes that it is not the surface stress, but the surface energy, measured via the surface tension, that is the controlling parameter.

Thank you for this summary, which is perfectly to the point.

The manuscript is well written and structured, but appears rather as a PRB than a body of work that would warrant publication in a journal for the broad general audience.

Our topic, the impact of capillarity for strength at the nanoscale, is relevant since 1.) the microscopic origin of the extremely high strength of metal nanostructures – a technologically important effect – remains under discussion and since 2.) capillary forces have been suggested to impose a lower limit on the stable size of crystals – this concerns the nanosciences at a fundamental level. We therefore see our paper as highly relevant to general audiences from material science, chemistry, solid-state physics, and from the full range of disciplines that are concerned with nanoscale devices or with applications of nano objects (catalysis, biology medicine).

The key novelty which we present to these audiences is twofold: Our argument on driving forces rejects a standard concept in the field and instead proposes an alternative picture, connecting to such fundamental thermodynamic quantities as the capillary terms surface tension and surface stress. Our experiments are original and they support our conclusions in what we see as an exceptionally compelling manner.

Our topic is of high current interest: the key references for surface-induced spontaneous deformation [4-10] and tension-compression anisotropy [6-8,11-14] together have accumulated 973 ISI citations

We would be pleased to learn that this fundamental and original science can be seen as suitable for Nature Communications.

In addition, it seems that some proposed ideas for the origins of the specific mechanical response of nanoporous gold (npAU) have been postulated by some of the authors and are now rejected. That means, the authors seem to contradict their earlier propositions. There is a priori nothing wrong with this, but NatComm does not seem to be the medium to discuss and address such matters. I would be more suitable in a specialized journal.

Our manuscript points towards misconceptions in the prior literature. The original postulate is not from the present authors – it can be traced back to earlier work by others (starting with Ref [6]) and to current claims by several groups, as is documented by citations in the Introduction. We consider it good scientific practice to not blame these misconceptions entirely on others but to openly admit that we supported them in the past.

The science reported in our manuscript is firmly connected to prior research, see our above remark on citations of the key references. We trust that any journal, and specifically a high-impact journal such as Nature Communications, will see this connection as an asset, not a liability. Whether or not that prior research contains contributions by the authors can be of no relevance – the key issue is, does the new work present original science and relevant progress. Certainly the reviewer will agree that their comment does not impair the significance of our manuscript.

The theory part is elaborate, and does present the motivation for the experiments in a concise way.

Thanks for these favorable remarks.

There are numerous steps that include quite some simplifications, which later seem to not matter, as the experiments are in agreement with the predictions of equation 12.

Thanks for addressing this. May we point out that good scientific practice requires that the simplifications (which are part of any comprehensible derivation) are documented, so that the reader can assess the range of applicability of the derivation. This supports our decision to maintain the Theory section, even in front of the different opinion of Reviewer #2.

For example, I find the picture of a dislocation gliding in the nanowire somewhat contrived.

In our perception, dislocation glide is the obvious and simplest picture – crystals at room temperature deform by dislocation glide. It is firmly established that this mechanism is also dominant in nanostructures:

Many molecular dynamics studies from the past decade support this view, examples are our Ref [13] for nanowires and, with specific attention to NPG, our Refs [7,8].

In experiment, high-resolution transmission electron microscopy has demonstrated dislocations in plastically deformed NPG, see Dou and Derby, *Phil. Mag.* 91 (2011) 1070 and in-situ during nanoindentation, see Sun et al., *Microsc Res Tech* 72 (2009) 232. Furthermore, experimental electron backscatter diffraction studies of NPG show the formation of a mosaic structure during progressive deformation, again evidence for lattice dislocation activity (Ref [53]).

First of all, I am not aware of any studies that actually have revealed the presence of any mobile full dislocation segments at such small length scales.

Agreed, there are good reasons to expect prevalently *partial* dislocation activity, see for instance Chen et al, *Science* 300, 1275 (2003). Yet the simulations in Refs [13] and [8] show both types of defects active, full and partial dislocations. Note that Ref [8] shows this in NPG for ligament sizes as small as 2 to 3 nm.

At no point does the paper's message depend on *full* dislocations as carriers of deformation, see below.

Secondly, dislocations in Au nanowires are mostly partials, where the nucleation stress is the rate limiting step for plasticity. As such, the discussion around the PK force and some dislocation mobility/traction force in the wires cannot be the relevant measure.

Surely the reviewer agrees that Peach-Köhler (PK) forces act irrespective of whether a dislocation is full or partial. In that respect, our argument applies independent of the nature of the dislocations.

We are in fact not aware that nucleation has been confirmed as the rate-controlling step in plastic flow of nanoscale structures. For instance, the pronounced strain hardening and the dislocation accumulation in NPG suggest dislocation interaction rather than nucleation as rate controlling, see Ref [8] and references therein. Pre-existing dislocations in as-prepared NPG (which need no nucleation) are documented in Refs [8,9]. The experimental strain rate sensitivity of NPG, Ref [53], is indeed very small in the initial stages of plastic deformation, which does rule *in* dislocation nucleation as rate controlling. Yet, a pronounced strain rate sensitivity develops beyond about engineering strain 0.2 (that is, in the strain regime explored by our experiments). This implies small activation volume, suggesting again dislocation interaction rather than nucleation as the rate controlling step.

In fact, our argument would hold even if nucleation were rate controlling: A simple approach to dislocation nucleation analyzes the energy barrier to be overcome by thermal activation in forming the critical nucleus. The energy of the nucleus contains contributions from the stacking fault and from its bounding partial dislocation ring. The yield stress enters the argument through the work which is done against the PK forces in creating the critical dislocation ring. Our discussion examines the PK forces; it is therefore immediately transferable to dislocation nucleation. The required external stresses are enhanced or diminished in the same way as for dislocation glide against dissipative forces as considered in our discussion. In

other words, the conclusions of our study stand even if dislocation nucleation were rate controlling.

As the most important and decisive argument, may we point out that the agreement between experiment and theory confirms that the suggested impact of the surface-induced stress – whether it acts during dislocation glide or during dislocation nucleation – is simply not the relevant issue.

The experiments are quite innovative, and interesting.

Thank you for this favorable comment.

There are, however, several aspects that remain unclear in the manuscript. From figure 4a, I understand that the plastic strain is more than 50%. The regime in which equation 12 fits the data is for an E_{SHE} between 0.3 and 1.3 V. This is the strain regime of somewhere from 20 to 45%. This is a very high amount of plastic strain, and given the structure shown in figure 3b, I would expect that the npAu is compacted during compression. As the authors state, this would reduce the net surface area during the mechanical experiment.

Yes, the reviewer's comment is perfectly to the point. Two remarks show that this change in microstructure does not impair our argument but may, in fact, even strengthen it:

Firstly, our (original and unprecedented) experiments monitoring the evolution of surface area, Fig S3 in the Supporting Online Material, show that its reduction is in the order of only 10% up to 45% plastic strain. Qualitatively, therefore, the microstructure remains consistent with the one at the start of the experiment.

Secondly, and as we discuss in detail, our figures 6a, 5b and S2c show that the predictions apply irrespective of the strain. Even though the microstructure changes, our theory holds. This emphasizes its transferability and, thereby, its relevance in general.

What is going on in the initial part of the stress-strain regime? There are ca. 20% of plastic strain that do not seem to be captured very well by the model. This is way beyond the elastic regime, and I do not see any plausible reason for the strong deviations between the prediction and the actual data. Actually, one would need to ask the question what makes the strain regime beyond 20% special to allow agreement with equation 12? Given the strong conclusions made by the authors, I am somewhat concerned about this fact.

The suggestion of “deviations” is not appropriate. The model was simply not tested against the initial stress-strain behavior, this does NOT imply failure of the model to capture the experiment.

Note, incidentally, that Fig S2a verifies the model in the wider strain regime, 12% - 52%.

Here is why we excluded the early stages of deformation:

Firstly, any mechanical test, and specifically compression testing of small samples, faces the technical issue that the “settling” of the sample into the load axis introduces artifacts at small strain. The data obtained for larger strains, beyond that settling stage, is therefore qualitatively more reliable.

Secondly, the early stages of deformation of nanoporous gold represent an extended elastic-plastic transition regime, well documented in the literature. Since our theory focuses on plasticity, the transition regime is not appropriate for testing the theory.

Thirdly, the mechanical tests (in the early stage of deformation) at constant electrode potential in the regime of “clean surface” provide a baseline by which the reader can assess the stability of the stress-strain response and the significance of potential-induced stress variations in the later stages.

In second paragraph of Section III we explain our strategy by stating that “The potential (...) was kept at 1.0V up to 20% engineering strain, establishing a reference for deformation at constant potential.”

In the discussion, the authors raise again the topic of a tension-compression asymmetry. Their experiments are done in compression, and not in tension. In order to justify the reoccurring focus on this matter, the

authors should also include tension data.

The discussion refers to tension-compression asymmetry in two places, each of which links to its prediction by the theory. This prediction is 1.) part of the state of the art (see above) and 2.) explained in more detail by the added passage in paragraph 3 of the Introduction (see also above).

We believe that the experimental compression test data in our manuscript are cutting edge. It would be absolutely fantastic to be able to even present similar experiments in tension. Yet, this is simply not doable at the present state of the art.

Meaningful in-electrolyte tests of nanoscale deformation behavior have only ever been achieved with NPG; transferring them to any other geometry would be a challenge even if (and that is not granted) stable tensile plastic flow could be achieved. Tension experiments on NPG (reported, specifically, by the groups of John Balk and of Karl Sieradzki) invariably show brittle failure in the elastic regime. Ref [27] exemplifies that the failure stress is here related to fracture mechanics concepts such as local stress concentrations and the statistics of flaws; it has no simple relation to our issues of plastic yield or flow. Furthermore, these experiments do not afford a probe for extended regimes of plasticity, as is required for investigating reversible changes in flow stress in response to the environment.

Note, however, that the molecular dynamics simulation results for NPG in Ref [7] compare compression and tension deformation, documenting a strong tension-compression asymmetry.

The bottom line is, yes, experimental tension data would be most desirable, but no, at present they are strictly not an option for the experimentalist. Our paper presents intelligent experiments in compression that still advance our understanding; we consider that as an asset.

As it stands, there is no evidence or data that would allow clarifying the origin of this asymmetry.

This point has already been addressed in the discussion with Reviewer #2. In short:

As we clearly state above and in the Introduction, the notion that a tension-compression asymmetry in the mechanical behavior of small structures originates in capillary forces has been proposed in earlier experimental [14-17] and simulation [6,11-13] work and is widely embraced in the literature. Our manuscript does NOT claim to invent that widely held view.

What we do show conclusively is: 1.) the surfaces do not – as is commonly believed – affect the mechanical behavior through the *surface stress*. 2.) we present strong and compelling evidence that *surface tension* does affect the mechanical behaviour. 3.) that latter effect agrees precisely with a theory that naturally and forcefully implies a tension-compression asymmetry (which originates in capillary forces). This also means that the well-established (by others) tension-compression asymmetry has a different origin than what the literature so far suggests. Reviewer #3 actually acknowledges this, our strategy, in the first paragraph of their comment.

We appreciate that the link between capillarity and the asymmetry may not have been sufficiently emphasized in the paper so far. Passages were added in the introduction, as already discussed. In the revised manuscript, the following passages emphasize the point and bring it to the immediate attention of the readers (but please recall that this is reference to the state-of-the-art, and not an original message of our paper):

Section I, third paragraph, “Zero creep experiments thus exemplify that tension-compression asymmetry results from the action of capillarity: creep is arrested by tensile stress but would be accelerated by a compressive stress of same magnitude.”

Section II, passage immediately after equation (8): “..., Eq 8 suggests strengthening in tension yet weakening in compression, in other words, a tension-compression asymmetry of the contribution of the surface

to stresses in small-scale plasticity.”

Section IV, 2nd sentence in Subsection Conservative versus dissipative processes: “The predicted change is positive, suggesting a tension-compression asymmetry with strengthening in tension and weakening in compression.”

In general, the discussion is probably a bit long. After all, the theory part already covered many of the raised aspects, and the experimental section makes most statements clear.

Thank you for the favorable assessment of the Experimental section.

Major parts of the discussion, and specifically the passages on the potential of zero charge and on the role of surface roughness, emerged from the discussion with the reviewers in the earlier review round. We feel that this enriched and clarified the message of the paper and so we are reluctant to remove arguments from the discussion.

In the revised manuscript, we have tentatively added subsection headers that make the structure of the arguments more immediately obvious. We believe that this substantially enhances the appeal and readability of this part of our manuscript.

Appendix: Excerpts from two references, see discussion with Reviewer #2

(citations refer to the bibliographies of the sources).

Ref. 9: “We report a macroscopic shrinkage by up to 30 vol% during electrochemical dealloying of Ag-Au. Since the original crystal lattice is maintained during the process, we suggest that the formation of nanoporous gold in our experiments is accompanied by the creation of a large number of lattice defects and by local plastic deformation.”

and later: “We shall now speculate on additional processes that could rationalize the experimental finding. As diffusion and elastic distortion can be ruled out (cf. the insignificant change in lattice parameter), it appears that plastic deformation needs to be taken into consideration. While this notion is supported by our observation of an increasing defect density at higher ED, correlating with the increasing shrinkage, none of the processes below has been confirmed. The idea of ligament shear touches on an issue of great interest in nanomechanics: In small cylindrical bodies, the surface-induced stress is not hydrostatic [15], and it has been suggested that this may cause spontaneous shear of Au ligaments [16]. Dislocation-mediated shear would require dislocation nucleation, which is an unresolved issue in nanostructures [17], and simulations of tensile loading suggest a transition from dislocation nucleation to homogeneous slip at a ligament diameter of ~1.5 nm [18]. In any case, surface stress-induced compressive yielding is expected to occur only in ligaments with diameters of several nanometers or less [16]. Although this diameter is rather smaller than that observed ex situ (Fig. 1), it is conceivable that such small diameter ligaments exist as a transient state, subject to rapid coarsening.”

Ref. 10: “The theme of plastic yielding under the action of the capillary forces is relevant for two aspects of our results, namely the shrinkage during reduction, Figure 1, and the irreversible deformation at very large stress and strain amplitude, Figure 4c. The process has in fact been invoked to explain the irreversible shrinkage during dealloying at large overpotential and the associated introduction of lattice defects, cf. ref 23. The same reference also explains why coarsening of the ligament structure of dealloyed np metals is not expected to result in shrinkage. The plastic yielding of np metal upon variation of the surface stress has recently been directly demonstrated in a molecular dynamics study.³² This supports the speculation that the irreversible length change in Figure 4c is the signature of plastic deformation. However, we tend to leave the issue open since (i), the structure and the stress state are quite complicated, and (ii) the absolute mean value of f for the oxygen-covered Au surfaces is unknown, so that it is not clear whether the net stress is compressive or tensile.”

Reviewers' comments:

Reviewer #2 (Remarks to the Author):

I am satisfied with the changes made to the original manuscript that appear in the revised manuscript. The authors have addressed my earlier concerns by modifying some of the introductory remarks, and I am pleased to be able to strongly recommend publication in Nature Communications.

Reviewer #3 (Remarks to the Author):

Obviously, the authors have the clear opinion that their work discovers important and fundamentally new science by conducting, as they write, original science. Even though I clearly value their work as high quality research, the manuscript itself does not do the job in outlining the importance. Since I am not directly from the npAu community, I leave the final judgement to the editorial office. I am still of the opinion that the work is excellent, but do not see the general fundamental importance. Whilst the authors make strong statements about the importance of their work in the response letter, I do not get the impression that there was a major effort made in meeting the reviewers' concerns. As a matter of fact, I find some of the responses quite unusual, as they engage in an argument without trying to improve the manuscript accordingly. Please bear in mind that I am solely evaluating the work and try to make the best recommendation possible to the editorial office. The tone of the authors is somewhat aggressive and not very often seen. I will only comment on a few responses made by the authors, and leave the rest to the editorial office.

In my comment about the PK force, I was questioning whether such an approach is fundamentally valid. The reason for this is the thought that the flow stress will be dictated by the nucleation stress for partials, and not by any mobile dislocation segments that are travelling through some lattice with any kind of barriers. The authors put forward that there is plenty of evidence of dislocation plasticity in npAu. Sure, deformation is going to be mediated by dislocations, something I was not questioning. However, looking at all the references used to respond to my comment, I find that there indeed is a lot of twinning observed – as one expects for partial dislocation activity. There does not seem to be any significant dislocation structure – as also expected for such length scales. The simulations should not serve as a very good reference, as they are probing athermal plasticity. Since the authors are that careful in responding to every sentence of my comment, I also want to point out that I never claimed the authors are reporting that their observations depend on full dislocations. Still, if plasticity is governed by nucleation of partials, as is the case in gold nanowires, how can the PK force then be a relevant measure?

Later on in their response, the authors first write that "qualitatively, therefore, the microstructure remains consistent with the one at the start of the experiments". In the following paragraph, they then write that the predictions apply irrespective of strain, meaning they do so even though the microstructure does change. I am not quite sure I understand this.

The response on my comment about why the experiment is best described for higher strains, the authors write that the initial plastic regime is not very reliable in such tests. I tend to agree, but 20% strain is quite a lot. I checked the literature and found that one of the authors has published work on small-scale testing of npAu pillars. Such an experiment may be seen as the worst in terms of boundary conditions and early stress-strain behavior. Yet, there is a well-developed flow regime at, say, 5%. In the present work, the authors use mm sized samples, so the elastic-plastic transition should be well defined.

In the response, it came to my attention that npAu actually is brittle in tension. This does not come out at all in the manuscript. As such, what is then the ground of comparison, if there is plasticity in compression, but none in tension? Certainly, the cited MD work will not be a solid ground for an answer.

When I ask about the tension-compression asymmetry, and the fact that the manuscript does not really give an origin to this behavior, the author's reply that they are not claiming to have invented this view. Note that I did not write that they did.

Reply to the reviewers' comments

We are most grateful to all reviewers for their careful and precise comments and reply in detail below. The passages in red summarize all revisions. Added references are listed at the end of this reply letter.

Reviewer #2 (Remarks to the Author):

I am satisfied with the changes made to the original manuscript that appear in the revised manuscript. The authors have addressed my earlier concerns by modifying some of the introductory remarks, and I am pleased to be able to strongly recommend publication in Nature Communications.

Thank you very much for this positive and welcome comment!

Reviewer #3 (Remarks to the Author):

Obviously, the authors have the clear opinion that their work discovers important and fundamentally new science by conducting, as they write, original science. Even though I clearly value their work as high quality research, the manuscript itself does not do the job in outlining the importance. Since I am not directly from the npAu community, I leave the final judgement to the editorial office. I am still of the opinion that the work is excellent, but do not see the general fundamental importance. Whilst the authors make strong statements about the importance of their work in the response letter, I do not get the impression that there was a major effort made in meeting the reviewers' concerns. As a matter of fact, I find some of the responses quite unusual, as they engage in an argument without trying to improve the manuscript accordingly. Please bear in mind that I am solely evaluating the work and try to make the best recommendation possible to the editorial office. The tone of the authors is somewhat aggressive and not very often seen.

We have the highest appreciation for the reviewer's effort in reviewing the manuscript and for the purposeful as well as helpful technical comments. The reply was intended as a scientific debate in precise and impartial wording. Our sincerest apologies if the reply appeared aggressive; this was in no way intended. The points of our rebuttal letter are now addressed in revisions to the manuscript.

I will only comment on a few responses made by the authors, and leave the rest to the editorial office.

In my comment about the PK force, I was questioning whether such an approach is fundamentally valid. The reason for this is the thought that the flow stress will be dictated by the nucleation stress for partials, and not by any mobile dislocation segments that are travelling through some lattice with any kind of barriers. The authors put forward that there is plenty of evidence of dislocation plasticity in npAu. Sure, deformation is going to be mediated by dislocations, something I was not questioning. However, looking at all the references used to respond to my comment, I find that there indeed is a lot of twinning observed – as one expects for partial dislocation activity. There does not seem to be any significant dislocation structure – as also expected for such length scales. The simulations should not serve as a very good reference, as they are probing athermal plasticity. Since the authors are that careful in responding to every sentence of my comment, I also want to point out that I never claimed the authors are reporting that their observations depend on full dislocations. Still, if plasticity is governed by nucleation of partials, as is the case in gold nanowires, how can the PK force then be a relevant measure?

We appreciate the relevance of dislocation nucleation. Let us try to put the argument briefly into perspective:

- Our work for the first time affords an experimental discrimination between the role of surface stress and of surface tension in affecting the strength of nanomaterials.

- Based on the experimental observations, our key proposition is that surface stress is not the central relevant parameter for strength and stability of small structures.
- This proposition rests on experiments at different electrode potential, and on their analysis by comparing the observed variation of flow stress to the known variations of surface stress and surface tension.
- Our conclusion does not rely on assumptions for the acting microscopic mechanisms, and specifically it does not rely on our consideration of PK forces – that argument in our text simply exemplifies why the discussion of surface stress in previous work may indeed not be forceful.
- Scenarios where dislocation nucleation controls the strength of small structures provide another example of a microscopic mechanism. Here again, the discussion of the role of surface stress in previous work is not forceful (see below). Bringing up that point is not necessary for supporting our conclusions, but it does strengthen them and at the same time widens the scope of our discussion.

With this in mind we address dislocation nucleation in the revised manuscript. In the interest of conciseness and comprehensibility the modifications are kept as brief as possible. We now reply in detail:

“simulations should not serve as a very good reference”: Thank you for addressing the limitations of the computer simulations. This emphasizes the need for experiment, supporting the relevance of our study.

“the flow stress will be dictated not by any mobile dislocation segments that are travelling through some lattice with any kind of barriers”: The state of the art suggests a different view. Figure 7 of Ref [8] shows the dislocation density in NPG to increase continuously from $5 \times 10^{14} \text{m}^{-2}$ before the onset of compression to $4 \times 10^{17} \text{m}^{-2}$ (20% of which are full dislocations) at true strain 0.8. Experimentally, the dislocation content in deformed NPG has been directly imaged by transmission electron microscopy [39] and indirectly evidenced by the formation of a mosaic structure [38]. Furthermore, matching finite element simulation to the experimental strain hardening of NPG requires Taylor work hardening in the constitutive law [40]. Each of these observations suggests that barriers to dislocation motion are relevant for the deformation of NPG. This is one motivation why twinning is not in the foreground of our considerations.

At the beginning of the 3rd paragraph, right column, page 3, we added (striving for brevity; see also below for the added remark on the microstructure evolution):

While dislocations from a stable or increasing population, for instance sustained from single-arm sources [37], may carry the plasticity and control the strength of small structures including NPG [8,38-40], nanowires may be dislocation-starved

The reviewer questions the validity of our PK stress argument since “the flow stress will be dictated by the nucleation stress for partials” and “there indeed is a lot of twinning observed – as one expects for partial dislocation activity”. In our understanding, the PK forces are equally relevant in nucleation (but see also our next point) as in dislocation travel. Twins are formed and propagated by the movement of partial dislocations and PK forces determine the interaction of the resolved shear stress with these processes. This is the origin of the term $\pi r^2 \sigma b$ in Eq 20.5 for the energetics of dislocation nucleation in the book by Anderson, Hirth and Lothe, our Ref [36]], which is the standard textbook in the field. Our considerations therefore do not rule out twinning, instead they naturally include that process. The spontaneous shear over an entire cross-sectional plane of the nanowire – an alternative microscopic mechanism for twinning – will interact with the resolved shear stress in exactly the same way (same work of deformation, hence same mean acting stress) as if a partial dislocation had crossed the entire cross-sectional glide plane. Again, the consideration of PK forces gives the relevant interaction term.

At the end of the paragraph discussing Peach-Köhler forces we have added (with reference to the relevant chapters of Ref [36]):

These forces act analogously on full dislocations and on partial dislocations that propagate a stacking fault or a twin. The lattice instability of nanowires at small size takes the form of twinning, so that the entire cross-section is sheared by the Burgers vector of a partial dislocation. The work against the acting stresses is again governed by the area integral of the traction [36], which vanishes.

“if plasticity is governed by nucleation of partials, as is the case in gold nanowires, how can the PK force then be a relevant measure?” The impact of the capillary forces for dislocation plasticity is less obvious when dislocations nucleate homogeneously at an extremely high stress: The critical nucleus is then extremely small and so it may selectively probe a *local* stress state. For metal surfaces with positive surface stress the stress state is compressive in the surface region but tensile in the bulk. It appears established [6,13,36,41] that dislocations nucleate at the surface. Ref [6] nonetheless discusses their MD results for nanowires with 1.5nm diameter in keeping with our Eq 9 (which relies on the bulk stress only), acknowledging the opposite-signed stresses but ignoring them in their conclusions. The issue is critically discussed in Ref [13], but not resolved. As Ref [13] confirms in their outlook, the observations so far remain dominated by computer simulation results and experiments on highly reproducible materials systems are urgently needed. These considerations STRENGTHEN the key proposition of our work: The role of surface stress for small-scale plasticity, as it is discussed in the previous literature, is in need of a dedicated study by meaningful and reproducible experiments. Our study provides such experiments along with a discussion that does not require assumptions on the detailed microscopic origin (e.g. dislocation nucleation or dislocation interaction) of the strength. These experiments turn out to qualify surface stress as irrelevant.

A note on nucleation has been included (page 3, column 2, third paragraph):

... , nanowires may be dislocation-starved and their strength controlled by dislocation nucleation [6,13,41]. As nucleation is favoured at *free surfaces* of bulk materials [36] and nanowires [6,13,41], the nucleation events then do not selectively probe the surface-induced bulk stress that leads to Eq 9 but they are affected by the large and opposite-signed stresses in the surface regions. This emphasizes that Eq 9 is not forceful and that the proposed impact of surface stress for the strength in small-scale plasticity requires experimental verification.

We have also reworded passages saying that our theory “rejects” surface stress and we instead now write in the abstract: Our theory qualifies the suggested impact of surface stress as not forceful and instead predicts a significant contribution of the surface energy, as measured by the surface tension.

in the first paragraph of Section IV: Our theory finds no forceful argument for a significant impact of surface stress on the plastic flow of nanowires.

Later on in their response, the authors first write that “qualitatively, therefore, the microstructure remains consistent with the one at the start of the experiments”. In the following paragraph, they then write that the predictions apply irrespective of strain, meaning they do so even though the microstructure does change. I am not quite sure I understand this.

Our apologies, we had focused the reply on the geometry of the nanoscale ligament network, ignoring the dislocation content. The reviewer correctly points out that this latter feature is also part of the microstructure. As we said, the findings for the variation of surface area with strain imply that the **network structure** does not change qualitatively. By contrast, the substantial strain hardening suggests that the **dislocation content** increases – see the discussion in our Refs [8,38,40].

In order to communicate this more clearly to the readers, we have added (last sentence of section III):

Thus, besides densifying the ligament network [8, 38], the plastic compression changes the microstructure by reducing the ligament aspect ratio, which decreases the net surface area. Furthermore, previous experiment [38] and atomistic [8] as well as continuum simulation [40] suggest that compression also enhances the dislocation density.

The response on my comment about why the experiment is best described for higher strains, the authors write that the initial plastic regime is not very reliable in such tests. I tend to agree, but 20% strain is quite a lot. I checked the literature and found that one of the authors has published work on small-scale testing of npAu pillars. Such an experiment may be seen as the worst in terms of boundary conditions and early stress-strain behavior. Yet, there is a well-developed flow regime at, say, 5%. In the present work, the authors use mm sized samples, so the elastic-plastic transition should be well defined.

Additional experiments, starting at 4% strain, are now presented in the revised Supporting Online Material (SOM) and referenced in the main text. The experimental observations at ALL strains – including the new data for the early stages of deformation – are consistent by sign and trend with our predictions for surface tension as the relevant coupling parameter, and they are inconsistent with the alternative parameter surface stress.

The second-last paragraph of Section III now reads:

Results of additional in situ compression tests, Fig S2 in the Supporting Online Material (SOM), confirm that the sign-inversion of the flow-stress potential response is recovered when scanning twice through E_{zc} , and that at all strains (down to values as small as 4%) the sign of the response is consistent with Eq 12 and with the prediction of the " $\Delta\gamma$ " graph of Fig 2. Thus, all experiments support surface tension as the relevant capillary force. The stronger and oppositely-signed response that would indicate surface stress as relevant (see the " Δf "-graph in Fig 2) is not supported by the experiment.

The SOM now shows the following modified Fig S2, including the new results in its parts b) – d):

Figure S2: In situ compression tests for NPG at constant engineering strain rate of 10^{-5} s^{-1} but with different potential step protocols. **a)** Test for a sample with ligament size $L=40$ nm in 1 M HClO_4 , with potential steps exploring anodic (positive-going) and reversed cathodic directions. Red: graph of stress σ versus strain ϵ ; blue: electrode potential E versus the standard hydrogen electrode (SHE). **b)** Test as in a), but for $L=30$ nm and in 0.5 M H_2SO_4 , here with potential jumps (in anodic direction) starting already at strain 4%. **c)** Flow stress-potential response, $\delta\sigma/\delta E$, versus the potential, $E - E_{zc}$, relative to the potential of zero charge, from the experiments in a) (blue circles) and in b) (red squares). Arrows show directions of the potential steps. Bold lines: predicted coupling strength near E_{zc} from Eq (12) of the main text, using the capacitance value $c=40 \text{ }\mu\text{F}/\text{cm}^2$ and ligament sizes as indicated in legend. **d)** Normalized flow stress response versus potential. Grey symbols: data from the in-situ tests compiled in Fig. 6 of the main text. Note the mutual consistency of all data sets, irrespective of ligament size and electrolyte.

All our experiments suggest a trend for the numerical magnitude of the response to be weaker in the earliest stages of the deformation. We advertise this to the readers in the revised manuscript. Figure S3b indicates that during this early stage the surface area of NPG varies only slowly with strain. Thus, taking into account the key assumptions in our theory, the weak flow-stress potential response can be expected.

This is now said in Section IV, 3rd paragraph of the passage “Aposing experiment and theory”, item *iii.*):

The decrease of the surface area of NPG with strain is least pronounced in the early stages of deformation, see Ref [8] and Fig S3. Consistent with our theory, the flow-stress potential response also tends to be less pronounced at small strain (see the data at most negative potential in Figs 5c, 6a, and S2d).

In the response, it came to my attention that npAu actually is brittle in tension. This does not come out at all in the manuscript. As such, what is then the ground of comparison, if there is plasticity in compression, but none in tension? Certainly, the cited MD work will not be a solid ground for an answer.

Here is a brief summary of our argument, as outlined in the manuscript: Surface-induced tension-compression asymmetry is an established notion in the field of small-scale plasticity. We use experiments on NPG to elucidate the physics behind that asymmetry, arguing against the role of surface stress and advertising the role of surface tension. While NPG is macroscopically brittle in tension, the insights obtained from studies of that material in compression promote our understanding of small-scale plasticity in general, including other materials that can be tested in tension. We hope that these thoughts reemphasize a leitmotif of our manuscript: The work is not guided by the intention to understand plasticity of NPG. Rather, that material is studied as a model system which provides unprecedented insights with relevance for small-scale plasticity in general.

In order to communicate the brittleness of NPG in tension more clearly to the readers and to emphasize the potential of NPG as a model material, the following passage has been added in the last paragraph of Section I:

The brittle failure of NPG in tension relates to fracture mechanics concepts such as the distribution of heterogeneities in the network structure [22]. By contrast, the material's excellent deformability in compression provides opportunities for probing the mechanisms and driving forces of yielding and plastic flow in small scale plasticity.

When I ask about the tension-compression asymmetry, and the fact that the manuscript does not really give an origin to this behavior, the author's reply that they are not claiming to have invented this view. Note that I did not write that they did.

In our previous revision we added text that explains the origin of the tension-compression asymmetry. The added passages were listed in the reply to reviewer #2, who had also addressed the point (and who was satisfied by our reaction). This text is again highlighted in the revised manuscript:

Section I, third paragraph, “The stress which is required for compensating the trend for contraction -- resulting in zero creep rate -- in a wire of radius r is tensile and of magnitude γ/r [15,16]. Zero creep experiments thus exemplify that tension-compression asymmetry results from the action of capillarity: creep is arrested by tensile stress but would be accelerated by a compressive stress of same magnitude.”

Section II, passage immediately after equation (8): “..., Eq 8 suggests strengthening in tension yet weakening in compression, in other words, a tension-compression asymmetry of the contribution of the surface to stresses in small-scale plasticity.”

Section IV, second sentence in Subsection entitled “Conservative versus dissipative processes”: “The predicted change is positive, suggesting a tension-compression asymmetry with strengthening in tension and weakening in compression.”

=====

The revision comprised the following additions to the reference list:

[22] N. Badwe, X. Chen and K. Sieradzki, *Mechanical properties of nanoporous gold in tension*. *Acta Mater.* 129:251–258, 2017.

[37] S. H. Oh, M. Legros, D. Kiener, and G. Dehm. *In situ observation of dislocation nucleation and escape in a submicrometre aluminium single crystal*. *Nature Mater.* 8:95–100, 2009.

[39] R. Dou and B. Derby. *Deformation mechanisms in gold nanowires and nanoporous gold*. *Phil. Mag.* 91:1070–1083, 2011.

[40] N. Huber, R. N. Viswanath, N. Mameka, J. Markmann, and J. Weissmüller. *Scaling laws of nanoporous metals under uniaxial compression*. *Acta Mater.* 67:252–265.

[41] E. Rabkin, H. S. Nam, and D. J. Srolovitz. *Atomistic simulation of the deformation of gold nanopillars*. *Acta Mater.* 55:2085–2099, 2007.

REVIEWERS' COMMENTS:

Reviewer #2 (Remarks to the Author):

Editorial Note: this reviewer provided comments to the Editors only.